# Effect of Surface Morphology Changes on Optical Properties of Silicon Nanowire Arrays

**DOI:** 10.3390/s22072454

**Published:** 2022-03-23

**Authors:** Shanshan Wang, Shujia Huang, Jijie Zhao

**Affiliations:** 1Electronic Information Engineering, Xi’an Technological University, Xi’an 710021, China; 2School of Optoelectronic Engineering, Xi’an Technological University, Xi’an 710021, China; h17389194199@163.com (S.H.); zhaojijie@st.xatu.edu.cn (J.Z.)

**Keywords:** silicon nanowires array, quantum size effect, quantum confinement effect, light absorption, AAO membranes

## Abstract

The optical properties of silicon nanowire arrays (SiNWs) are closely related to surface morphology due to quantum effects and quantum confinement effects of the existing semiconductor nanocrystal. In order to explore the influence of the diameters and distribution density of nanowires on the light absorption in the visible to near infrared band, we report the highly efficient method of multiple replication of versatile homogeneous Au films from porous anodic aluminum oxide (AAO) membranes by ion sputtering as etching catalysts; the monocrystalline silicon is etched along the growth templates in a fixed proportion chemical solution to form homogeneous ordered arrays of different morphology and distributions on the surface. In this system, we demonstrate that the synthesized nanostructure arrays can be tuned to exhibit different optical characteristics in the test wavelength range by adjusting the structural parameters of AAO membranes.

## 1. Introduction

An important application of silicon nanowire arrays (SiNWs) are silicon-based sensors, which mainly realize the sensing purpose by detecting changes in various substances caused by the small changes in nanowires’ interaction with foreign substances or fields [1,2,3]. An important challenge of advanced silicon-based sensors is adjusting the band range of light absorption to expand the flexibility in detection. The quantum confinement of light excitation carriers can widen the energy band inside SiNWs and improve the transition probability of stimulated radiation of the electrons [4,5]. Therefore, the energy band structure of nanomaterials shows the obvious size dependence. The above illustrates that the light absorption characteristics of materials in different wave bands can be realized by the precise control of nanowire growth.

At present, the more mature processes for the preparation of silicon nanowire arrays are laser ablation, thermal evaporation, chemical vapor deposition (CVD) with (vapor-liquid-solid) VLS mechanism and electrochemical deposition with (solution-liquid-solid) SLS mechanism. The main differences between several methods are the heating mode and major silicon source. Laser ablation is used to directly radiate the surface of material above the boiling point by using a high-energy laser as the heat source [6,7,8]. The thermal evaporation is used for heating to a high temperature of 1200 °C and vacuum conditions [9,10]. The sample prepared with CVD is heated in a quartz tube. Gold is commonly used as the catalyst in the CVD method because of its low temperature of eutectic formation and good chemical stability; the quartz tube is heated to the lowest eutectic temperature of Si/Au (about 363 °C) [11,12,13]. Meanwhile, solid-state silica powder is used as a silicon source in laser ablation. The silicon source of thermal evaporation is a mixed powder of high-purity Si and SiO_2_ with weight ratio of 1:1; the CVD method uses SiCl_4_ or SiH_4_ by high-temperature decomposition to provide silicon atoms. One of the most important advantages of the VLS is the flexibility of the approach, which can be adapted to different equipment and approaches. Another advantage is the possibility to obtain SiNWs with a diameter of about 10 nm and even below this limit [14]. However, the growth of SiNWs is catalyzed by metal droplets that are realized as a product of the melting of metal nanoparticles contaminated with silicon atoms [15]. The synthesis is still complex and the SiNWs have different orientations, making their implementation for real applications more complex [14].

Compared with the above three methods, the electrochemical method has attracted extensive attention due to the advantages of lower deposited temperature and no residual thermal stress in the synthetic materials, which enhances the bonding force between the substrate and the synthetic materials. However, the synthetic process of the materials is affected by many external experimental factors. Even in the above four methods with complicated processing, the fabricated SiNWs still have difficulty in presenting a homogeneous ordered and higher density structure.

In contrast, integrating the merits of the previous four methods, metal-assisted chemical etching (MACE) has the advantage of short cycles, mild reactions, large area growth and no metal contamination, and the implementation of this approach does not require high temperatures, vacuum environments and expensive equipment [16,17]. Traditional MACE places the silicon wafer into an AgNO_3_ solution for silver plating. The etching mechanism originates from the possibility for exploration of the initial position of Ag particles, whereby the silicon just covered with the Ag layer can be catalytically etched [18]. Due to the random distribution of silver particles on the silicon surface, the adjacent silicon nanowire is prone to clustering and the morphology of the array cannot be accurately controlled. More advanced syntheses are optimized to control SiNW diameter and planar arrangement by using single-step MACE coupled with lithography processes [14]. Nonetheless, photolithography is limited to small areas of about 10^4^ µm^2^.

The recent research has shown that the metal film attached to the surface of silicon substrates offers an etching template to control the diameter, spacing and density of SiNWs in the MACE. Leonardi [19] engineered a modified approach that surpasses the limit of the traditional MACE through the use of thin metal. Au or Ag layers of a few nanometers are deposited onto the silicon substrates by electron beam evaporators to obtain high crystalline quality SiNWs. The thickness of the metal film is required to be less than 10 nm, which requires accurate control of experimental parameters. Kim et al. developed Au/Ag bilayered metal meshes with regular hexagonal arrays of nanoholes that can be prepared from AAO membranes as a catalyst for wet-chemical etching of silicon. Au is inert against oxidative dissolution in a mixture solution of HF and H_2_O_2_; the upper Au layer in Au/Ag bilayered metal mesh prevents not only undesired structural disintegration of the underlying Ag layer but also tapering of SiNWs, still acting as a catalyst for H_2_O_2_ decomposition on its surface [20]. This view is consistent with our previous experimental observations. The silver film dissolves at higher temperature or longer etching time, resulting in structural damage. To overcome this problem, our approach devises single-layer Au films by ion sputtering as etching catalysts. Compared with the Au/Ag bilayered metal mesh, the complexity of the experiment is reduced, but the control requirements for the parameters of the Au film are improved.

The experimental system proposed in this paper is based on acidic anisotropic etching of silicon along with metal-assisted catalysis by means of nanostructured Au films formed by ion sputtering AAO membranes, overcoming drawbacks involved in traditional MACE. The diameter and distribution density of silicon nanowires can be controlled by adjusting the structural parameters of AAO membranes. Besides, we further test the optical properties of arrays with different diameters and distribution density. The abnormal optical performance caused by changes in the surface morphology of nanowire arrays expand their application prospects in the field of new optoelectronics detection.

## 2. Experiment

### 2.1. The Preparatory Work

The characteristic parameters of monocrystalline silicon affect the morphology of SiNWs formed by etching, which is related to the number of holes in the semiconductor. Generally, P-type monocrystalline silicon is more suitable for etching and N-type monocrystalline silicon usually requires illumination or a high electric field to stimulate the holes. Silicon wafer produced by Zhejiang Lijing photoelectric technology company is used in this experiment. The parameters are as follows: single polished P-type silicon wafer, 500 ± 10 μm in thickness, 100 ± 0.4 mm in diameter, <100> in crystal orientation, and 1–10 Ω·cm in resistivity.

The original silicon wafer is cut into 10 mm × 10 mm square pieces by a diamond knife, which is immersed in ethanol, 5%HF and ammonia solution successively, cleaned by ultrasound for 10 min and dried with high-purity nitrogen at each step. The contaminated layer on the surface of the silicon wafer has a great influence on the adhesion ability between substrate and Au films, as detailed later [21].

### 2.2. Formation of Au Films

AAO, which is made from 99.999% high-purity aluminum sheet with a series of complex processes, is a kind of porous hard membrane used for preparing various nanostructures [22]. AAO produced by Topmembranes Technology Co., Ltd., Shenzhen, China is used in the experiment. The AAO models are as follows: DP450-200S-50000, DP450-250S-50000, DP450-300S-50000 and DP450-350S-50000. The center distance of the two adjacent holes of the four samples is 450 nm, the thickness is 50,000 nm, the exterior diameter is 25 mm, and the pore diameters are 200 nm, 250 nm, 300 nm and 350 nm, respectively.

Four different types of AAO membranes are sputtered by ion sputtering coater (sbc-12, KYKY technology, Beijing, China). The thickness of the Au films formed by sputtering is an important factor in the formation of ordered arrays on the surface of silicon substrates. The qualified Au films can be obtained by accurately controlling the input current, sputtering time and sputtering number. The sputtering parameters are selected as follows: current 3 mA, time 110 s, sputtering twice under this condition [23]. The Au film formed in the experiment cannot be broken due to the surface tension of the solution, nor can the Au nanomeshes be blocked by Au particles due to the closure effect [24].

### 2.3. Transferring of Au Films

Double-channel AAO membranes are utilized as a sacrificial layer for replicating Au films by selective chemical dissolution of AAO from ultra-thin Au-coated AAO membranes [22]. The AAO covered with Au films are gently immersed into a 1.875 mol/L NaOH solution for about 10 min [25]. The Au films, replicating the morphology of AAO, are separated from the AAO by chemical dissolution of AAO in a 1.875 mol/L NaOH solution at room temperature. Owing to the surface energy of ultra-thin Au films, the Au films remain floating on the surface of the NaOH solution, and then are transferred to the cleaned silicon wafers. The Figure 1 shows that the ordered meshes pattern of Au films on the silicon substrates are observed clearly by a scanning electron microscope (SEM, SU1510, Hitachi, Tokyo, Japan). The single nanomesh diameter of desired Au films is measured to be standard size plus or minus 20 nm. This indicates that the size of Au films can be adjusted by adopting AAO with different structural parameters.

### 2.4. The Etching Process

The adhesive force between the Au films and the substrate is one of the important factors in the formation of ordered arrays in the etching process. There are many factors affecting adhesion. One is the cleanliness of the original silicon wafers, as mentioned earlier; another is the deposition rate of the Au films in the sputtering process. As the deposition rate rises, the looser is the Au film texture, and the worse the adhesive performance. However, Au films are too hard to move toward the substrate as a catalyst during etching. The above are experimental processes that need to be considered.

The silicon substrate covered with the spatial mesh ordering of Au films are placed in the etching solution ratio of H_2_O_2_:HF:H_2_O = 5:12:37 for 60 min [23]. The surface morphologies of various ordered structures were investigated via SEM; the geometry of the arrays is shown in Figure 2, demonstrating the excellent uniformity over large area that can be fabricated by this method. The diameters of the final nanowires formed by etching are smaller than that of the Au mesh formed by sputtering and are coupled to the etching time. The results prove that the Au films were used in the experiment not only as an etching catalyst, but also as a restricted template for nanowire growth.

## 3. Optical Characteristics Test

In order to study the potential application of our experimental results to advanced silicon-based sensors, the optical characteristics of the samples in a wavelength ranging from visible to near infrared were tested using a PerkinElmer UV/Vis/NIR Spectrometer (Lambda950, PerkinElmer, Waltham, MA, USA)—the sampling aperture of the integrating sphere of the spectrometer was 150 mm. The fundamental principle of the testing work is that light is collected with the sampling port and then scattered uniformly inside the integrating sphere after multiple reflections, thus obtaining high measurement accuracy. Figure 3 shows the comparison of optical characteristics for samples fabricated by the different types of AAO as a function of wavelength.

As depicted in Figure 3a, the transmissivity of the four samples is approximately zero because of the poor transmittance of light in the visible band to 1000 nm. From the 1000 to 1200 nm band, the transmissivity of samples prepared with AAO parameters of DP450-200s-50000, DP450-250s-50000, DP450-300s-50000 and DP 450-350s-50000 increased by 10.10%, 8.78%, 7.58% and 6.3%, respectively. Compared with polished silicon wafer, the transmissivity of the four samples with micro-nano structures on the surface was reduced significantly in the entire test band.

In Figure 3b, reflection measurement performed on before monocrystalline silicon and after ordered arrays show that the reflectivity over the test spectral range from 400 to 1200 nm is reduced dramatically by the four ordered arrays. The band gap of monocrystalline silicon is 1.12 eV, so the corresponding limit value of optical wavelength is about 1100 nm. The absorption coefficient of monocrystalline silicon to light with wavelength exceeding 1100 nm is very low and it is almost transparent in this band. In the 400–1150 nm band, the average reflectivity of samples prepared with AAO parameters of DP450-200s-50000, DP450-250s-50000, DP450-300s-50000 and DP 450-350s-50000 are 10.51%, 11.35%, 12.35%, and 13.29%, respectively. However, when the wavelength of the incident light exceeded 1150 nm, the reflectivity of the four samples increased by 11.37%, 10.36%, 9.34% and 8.21%, respectively. 

## 4. Discussion

The test results demonstrate that the light absorption properties at a fixed wavelength are related not only to the size of nanowires, but also to the distribution density of entire arrays. We quantitatively measured strong anti-reflection performance with path length enhancement by dispersing the incident light at various angles to the material [26], which effectively reduced the light energy reflection on the surface and enhanced the absorption of photons in the wide band. The eddy current generated on the surface of polished silicon wafers under the irradiation of electromagnetic waves seriously hinders the electromagnetic waves from free space into the material’s interior, which forms strong reflections on the semiconductor surface. Adjusting the size of the material to the nano level can not only effectively prevent the generation of eddy current, but also promote the absorption of specific wavelength light waves by controlling the morphology of SiNWs.

The incident light has the longest optical path in the array prepared by the DP450-200s-50000, among all ordered samples tested, and shows the lowest reflectivity in the 400–1150 nm band. The compact structure of arrays allows incident light to scatter and resonate more times between nanowires, which causes the enhancement of light absorption. It is possible that single nanowires act as effective scattering centers and the incident light is coupled into the film after multiple scattering [27].

It is shown in Figure 3 that the energy band structure of materials has an obvious size dependence. There is a possible reason for the reflectivity change in the near-infrared region. H_2_O_2_ is preferentially reduced in the vicinity of the Au film during the etching process. The holes generated by the reduction are diffused through the Au nanomeshes and injected into the silicon wafer below [28,29]. The hole concentration is the highest at the interface between the silicon wafer and Au film, so the silicon covered by Au film is etched the fastest. Moreover, the number of holes consumed is less than the number of injected holes at the interface. The surplus holes also diffuse to the surrounding area, which form nanopores on the side wall of the silicon wires. The larger the circumference of the Au film, the more holes are injected. The larger the diameter of the silicon nanowires, the more nanopores there are on the surface, and the stronger the adsorptive capacity of impurity. Electrons or holes bound to impurity levels can also cause light absorption. The presence of impurity levels may reduce the electron transition energy level.

Another explanation for the test results is that the small particles inside nanomaterials cause great surface tension, which induces lattice distortion and the lattice constant to decrease [30,31]. The smaller the particle size, the more atoms there are on the surface layer, indicating a serious lack of coordination among surface atoms. There are many dangling bonds that destroy the continuity of the lattice vibration wave inside the grain. The divergence and attenuation of electromagnetic waves by nanomaterials are completely different from bulk materials. The increase of unsaturated dangling bonds on the surface layer of nanomaterials lacks a single and preferred bond vibration mode, but there exists a wider distribution of chemical bond vibration modes, which is conducive to broadening the frequency range of absorbed electromagnetic waves. In addition, the interfacial polarization relaxation of particles can significantly improve the performance of absorbing electromagnetic waves. Leonardi et al. [32] point out that the innovative nanomaterial properties, as well as their increased high surface-to-volume ratio, make nanotechnology a strategic tool for surpassing the standard sensor limits.

Furthermore, scientists are concerned about nano oxide and nitride particles; they reveal that the distance between first proximity and second proximity becomes shorter, which causes the increase of the bond intrinsic vibration frequency of nanoparticles, resulting in the shift of the infrared absorptive band to a high wave number. Another view is that the plasma resonance of nanoparticles is also affected by the quantum size. The particle size can be adjusted to control the displacement of absorption peak frequency and absorb electromagnetic waves in a specific frequency band.

Nakanashi et al. prepared peryene nanocrystals by precipitation and found that the the absorption spectrum of perylene nanocrystals shifts towards blue with the decrease in particle size [33,34]. They analyze the reason as being the increase of surface area of nanocrystals results in the change of lattice; the increase in the number of surface molecules causes lattice relaxation in the nanocrystal’s interior. The Coulomb force between molecules becomes smaller and the energy band gap becomes larger, causing the blue shift. This view can also be used to explain our experimental results. Optical measurements performed on homogenously distributed ordered nanowire arrays exceed the 1150 nm band; the increasing amplitude of reflectivity decreases with the increase in nanowire diameter. The other possibility is that the blue shift of the absorption spectrum is caused by the electric field of the medium around the nanocrystal as it passes through its surface.

## 5. Conclusions

The application potential of silicon nanowire arrays for advanced silicon-based sensors lies in controlling its optical properties via the change of its surface morphology. We demonstrate a flexible and convenient approach to fabricate ordered SiNWs with different diameters and distribution densities by utilizing double-channel AAO membranes with different structural parameters. We further tested the optical properties of the uniformity over large area SiNWs of four different structures. The incident light has the longest optical path of the four samples and present the lowest reflectivity in the 400–1150 nm band. The compact structure of the array disperses incident light at various angles to the array, causing more scattering and resonance, which cannot be accomplished by polishing silicon wafers. However, when the wavelength of incident light exceeds 1150 nm, the rising amplitude of reflectivity decreases with the increase in nanowire diameter. This state can be explained by the model formula of the particle in the classical box, according to which the effective band gap is inversely proportional to *R*^2^. Part of the reason is the unique surface effect of nanomaterials: the smaller the particles are, the more dangling bonds there are on the surface. The increase in unsaturated dangling bonds gives rise to a wide distribution of chemical bond vibrating modes, which is conducive to broadening the frequency range of electromagnetic waves absorbed by materials. Another reason is that the interfacial polarization relaxation of nanoparticles and multiple scattering caused by light in the micro-nano structure array can significantly improve the performance of absorbing light waves. In addition, the test results of optical characteristics demonstrate that the energy band structure of nanowires presents the obvious size dependence. The particle size can be adjusted to control the displacement of absorption peak frequency and absorb electromagnetic waves in a specific frequency band. The homogeneous ordered arrays prepared by this method exhibit superior optical tunability that offers promising applications for improving the properties of advanced silicon-based sensors.

## Figures and Tables

**Figure 1 sensors-22-02454-f001:**
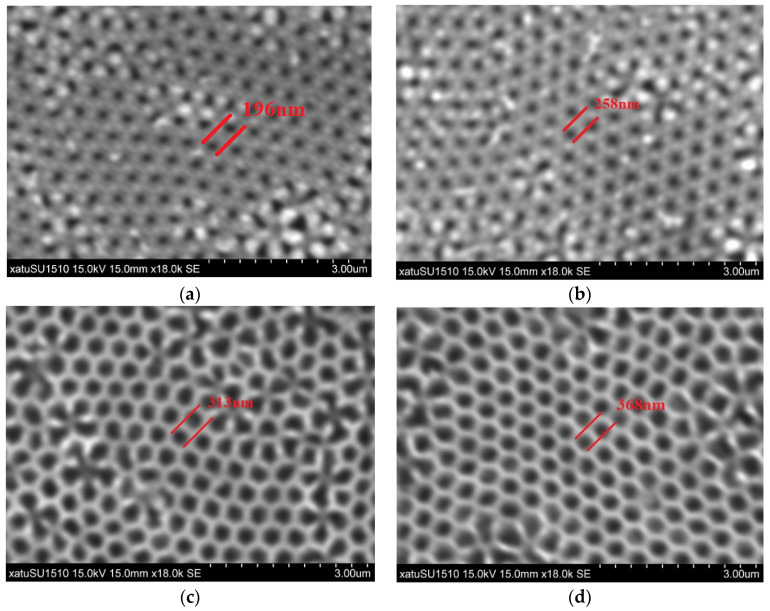
The SEM images of Au films are obtained by replicating the ultra-thin AAO membranes, while the AAO membranes were dissolved with 1.875 mol/L NaOH solution to form a sacrificial layer before SEM images were taken: (**a**) DP450-200S-50000; (**b**) DP450-250S-50000; (**c**) DP450-300S-50000; (**d**) DP450-350S-50000.

**Figure 2 sensors-22-02454-f002:**
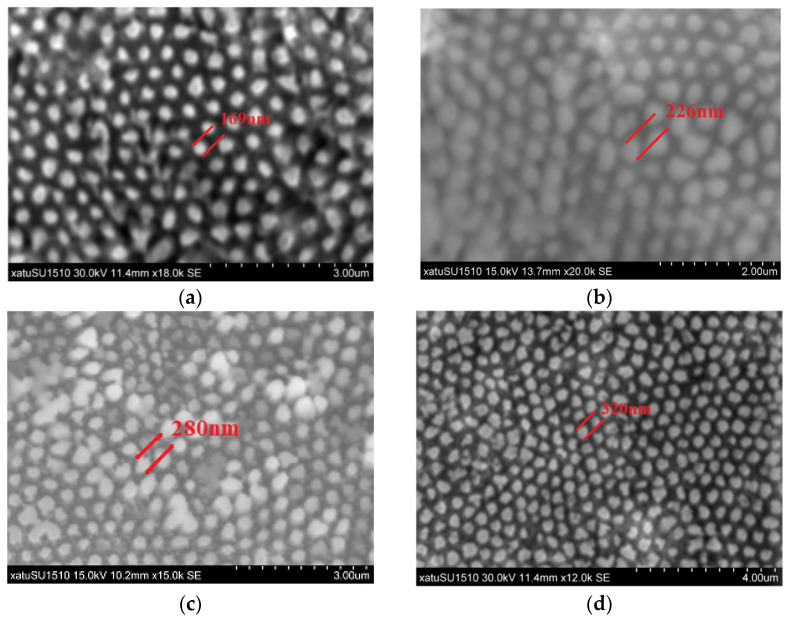
The SEM images of SiNWs fabricated using four different parameters of AAO membranes with the same sputtering parameters: (**a**) DP450-200S-50000; (**b**) DP450-250S-50000; (**c**) DP450-300S-50000; (**d**) DP450-350S-50000.

**Figure 3 sensors-22-02454-f003:**
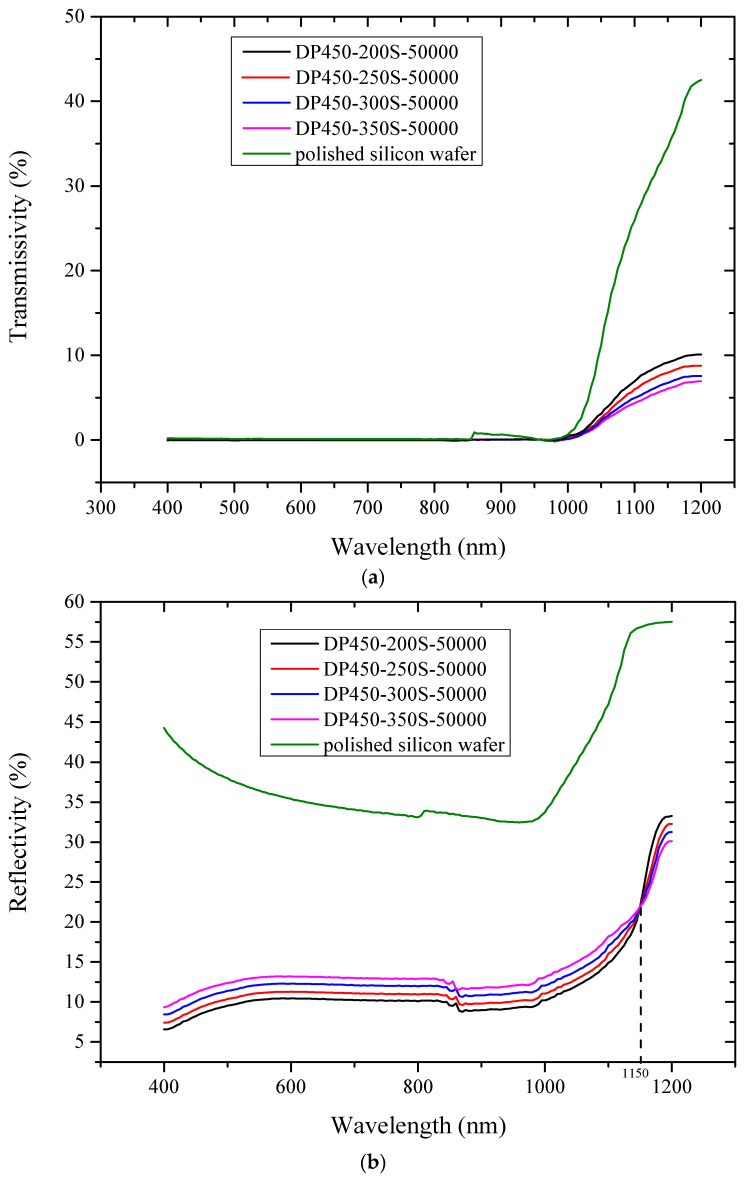
The comparison of optical characteristics for all samples fabricated from the different parameters of AAO: (**a**) the comparison of transmissivity for all samples at 400–1200 nm band; (**b**) the comparison of reflectivity for all samples at 400–1200 nm band.

## Data Availability

Not applicable.

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
