# Peer review of "Effect of Surface Morphology Changes on Optical Properties of Silicon Nanowire Arrays"

_sensors, 2022, doi:10.3390/s22072454_

Round 1

Reviewer 1 Report

The manuscript of S. Wang et al. is focused on the MACE fabrication of Si NWs using an AAO as mask for the gold film deposition followed by the reflectivity measurements. In particular, the authors focus their efforts on the optical study of the NWs envisioning further application as sensors. The topic even if not completely new is interesting seeing the relevance of novel method of Si NWs synthesis that can pave the route toward their industrial application. Despite that, I have some serious concern on the quantum confinement nature attributed to the reflectivity results and to the literature survey.

  1. MACE is not anymore just silver salt approach and already a lot of different route involving metal catalyst in one (all during the wet etch) or two step (i.e. metal deposited by a PVD approach) procedures exists. Even considering the sensor application of Si NWs a lot of works have been produced in these years. A brief discussion on other MACE approaches and on their comparison respect the one used by the authors should be inserted [Nanomaterials 2020, 10(5),966; https://doi.org/10.3390/nano10050966; Nanomaterials 2021, 11(2),383, https://doi.org/10.3390/nano11020383; Anal. Chim. Acta 2021, 1160, 338393 https://doi.org/10.1016/j.aca.2021.338393; ACS Nano 2011, 5, 4, 3222–3229 https://doi.org/10.1021/nn2003458 ].

  1. I have never seen the term quantum special effect. What is the meaning? Can you please at least provide a reference that use “quantum special effect” and what the authors refers to?

  1. It is well known in literature that for silicon to observe quantum confinement effect a size around few tens of nanometers or even less is required. Seeing the dimension of the Si NWs produced by this approach I am not convinced that the eq. used by the authors can be adopted (moreover, put a number on the eq.). Moreover, if quantum confinement is present a certain variation between each different sample characterized by different Si NW diameters should be observed (and doesn’t seem so). Indeed, maybe other effects can justify the reflectivity trend as defect formation, impurity contamination or even nano-porosity of the Si NWs that can determine the presence of nanocrystals (few nanometers) that are quantum confined. Without any other measure that can justify the quantum confinement this is not acceptable and as to be removed and if inserted as a possibility (i.e. nanoporous case) should be better explained.

Reviewer 2 Report

This work shows the effect of surface morphology changes on optical properties of silicon nanowires array. The authors demonstrate that the synthesized nanostructure arrays can be tuned to exhibit different optical characteristic in test wavelength range by adjusting the structural parameters of AAO membranes. This research may have potential applications in the field of new optoelectronic detection. In general, the structure of the manuscript is clear, and experimental data are abundant. The paper could be published after minor revision addressing the following points.

  1. In page 6, section IV: Discussion, please give the meaning of each parameter in the formula.
  2. Weather the nonuniformity of nanowires diameter have effect to opticalcharacteristics or not, and how to effect.
  3. There seems to be a mismatch between the reference number and the article, e.g. ref 26 seems not correct. Please check if the cited articles are correct.

Round 2

Reviewer 1 Report

The authors revised the manuscript. The paper is suitable for publication